# High Diversity of *Leptospira* Species Infecting Bats Captured in the Urabá Region (Antioquia-Colombia)

**DOI:** 10.3390/microorganisms9091897

**Published:** 2021-09-07

**Authors:** Fernando P. Monroy, Sergio Solari, Juan Álvaro Lopez, Piedad Agudelo-Flórez, Ronald Guillermo Peláez Sánchez

**Affiliations:** 1Department of Biological Sciences, Northern Arizona University, Flagstaff, AZ 86011, USA; 2Institute of Biology, University of Antioquia, Medellín 50010, Colombia; sergio.solari@udea.edu.co; 3Microbiology School, Primary Immunodeficiencies Group, University of Antioquia, Medellín 50010, Colombia; alvaro.lopez@udea.edu.co; 4Basic Science Research Group, Graduate School—CES University, Medellín 50021, Colombia; pagudelo@ces.edu.co (P.A.-F.); rpelaezp@ces.edu.co (R.G.P.S.)

**Keywords:** *Leptospira*, bats, Colombia, leptospirosis, species, type, 16S ribosomal gene

## Abstract

Leptospirosis is a globally distributed zoonotic disease caused by pathogenic bacteria of the genus *Leptospira*. This zoonotic disease affects humans, domestic animals and wild animals. Colombia is considered an endemic country for leptospirosis; Antioquia is the second department in Colombia, with the highest number of reported leptospirosis cases. Currently, many studies report bats as reservoirs of *Leptospira* spp. but the prevalence in these mammals is unknown. The goal of this study was to better understand the role of bats as reservoir hosts of *Leptospira* species and to evaluate the genetic diversity of circulating *Leptospira* species in Antioquia-Colombia. We captured 206 bats in the municipalities of Chigorodó (43 bats), Carepa (43 bats), Apartadó (39 bats), Turbo (40 bats), and Necoclí (41 bats) in the Urabá region (Antioquia-Colombia). Twenty bats tested positive for *Leptospira* spp. infection (20/206—9.70%) and the species of infected bats were *Carollia perspicillata*, *Dermanura rava*, *Glossophaga soricina*, *Molossus molossus*, *Artibeus planirostris*, and *Uroderma convexum.* These species have different feeding strategies such as frugivorous, insectivores, and nectarivores. The infecting *Leptospira* species identified were *Leptospira borgpetersenii* (3/20–15%), *Leptospira alexanderi* (2/20–10%), *Leptospira noguchii* (6/20–30%), *Leptospira interrogans* (3/20–15%), and *Leptospira kirschneri* (6/20–30%). Our results showed the importance of bats in the epidemiology, ecology, and evolution of *Leptospira* in this host-pathogen association. This is the first step in deciphering the role played by bats in the epidemiology of human leptospirosis in the endemic region of Urabá (Antioquia-Colombia).

## 1. Introduction

Leptospirosis is an emerging zoonotic disease caused by pathogenic bacteria of the genus *Leptospira* [1]. Previous studies have estimated that 1.03 million cases and 58,900 deaths occur due to leptospirosis worldwide annually [2]. Leptospirosis is considered a neglected disease, found mainly in the tropical regions of developing countries [3] and is now recognized as an emerging infectious disease due to large outbreaks in different regions of the world, which are associated with environmental disasters, and extreme climate change. In many endemic regions, severe forms of the disease, such as Weil’s disease and pulmonary hemorrhage syndrome have emerged as the leading cause of death [4]. Currently, about 65 genomic *Leptospira* species have been identified (NCBI database: https://www.ncbi.nlm.nih.gov/genome [accessed on 30 April 2021]), which are subdivided into four main clades according to the phylogenetic analysis of 1371 conserved genes: pathogens (P1), pathogens (P2), saprophytes (S1), and saprophytes (S2) [4,5]. Through serological classification about 300 *Leptospira* serovars have been described, which are grouped into approximately 30 serogroups and about 200 of these serovars have been considered pathogenic [6]. Colombia is an endemic country for leptospirosis with at least 500 cases every year [7]. Antioquia is the second department in Colombia with the highest number of confirmed cases of leptospirosis [7], with a seroprevalence close to 12.5% [8]. *Leptospira interrogans* and *L. santarosai* have been identified as the causative agents of this disease [9]. Therefore, this department in an important region in Colombia for the study of leptospirosis.

Rodents and dogs are often identified as potential sources of human infection, but other mammals have also been identified in the transmission cycle of leptospirosis [1]. Globally, various studies have explored the biological role of bats as reservoirs of zoonotic pathogens due to their ability to fly long distances and disperse pathogens (viruses [10], bacteria [11], parasites [12] and fungi [13]) through urine, saliva, and feces. Bats are the only true flying mammals, belonging to the order Chiroptera [14]. This order includes over 1400 different species in 21 families [15], which are scattered throughout the world, except Antarctica [16]. These mammals are oriented and hunt by echolocation [17]. Depending on the species they can feed on arthropods, fruits, pollen, fish, blood, and other vertebrates (carnivores) [10]. Some species can hibernate [18], form large colonies [19], migrate long distances [20], and have long lifespans (approximately 35 years) [21].

Bats have been identified worldwide as an important reservoir of different *Leptospira* species (*L. interrogans*, *L. borgpetersenii*, *L. kirschneri*, *L. fainei*) and their role in disease transmission, and spillover in the life cycle of this bacterium has yet to be defined [22]. Currently, more than 50 species of infected bats with *Leptospira* have been reported in different countries, including Peru [23], Brazil [24], Argentina [25], Australia [26], Comoros island and Madagascar [27], Reunion Island [28], Mayotte Island [29], Indonesia [30], Malaysia [31], Tanzania [32], Trinidad [33], Sudan [34], Democratic Republic of Congo [35], Africa [36], and Azerbaijan [37]. In Colombia, two studies have reported the presence of bats naturally infected with *Leptospira* [38,39]. Due to the above characteristics, bats could act as excellent spillover of *Leptospira* species to the environment, favoring contamination of water and soil, serving as a direct or indirect source of infection for other animals, which are the main reservoirs and disseminators of the bacteria. The objective of the present investigation was to detect *Leptospira* species infecting different bat species in the Urabá region (Antioquia-Colombia) and to evaluate the genetic diversity of the circulating *Leptospira* species. This information will illustrate the role of bats in the transmission cycle of human leptospirosis.

## 2. Materials and Methods

### 2.1. Ethical Considerations

This research was authorized by the National Authority of Environmental Licenses of Colombia (ANLA) according to resolution 0524 of 27 May 2014, which grants permission to collect wild species of biological diversity for non-commercial scientific research purposes. This research was endorsed by the Ministry of Environment and Sustainable Development of the Republic of Colombia.

### 2.2. Characteristics of the Capture Area of Specimens

Urabá is a geographical sub-region of Colombia; its name literally means freshwater gulf, due to the low salinity of the gulf’s waters, which is achieved by the mixture of seawater with large rivers flowing into the gulf. This region is surrounded by the Pacific Ocean to the west, and the Caribbean Sea to the northeast. The region is made up of eleven municipalities (Arboletes, San Juan de Urabá, San Pedro de Urabá, Necoclí, Apartado, Carepa, Chigorodó, Turbo, Mutatá, Murindó, and Vigía del Fuerte). With respect to its geographical characteristics; the disposition of its lands is of the plain type, Caribbean eco-region, surface of 11,664 km^2^, average altitude of 919 m above sea level, 659,266 inhabitants (10.3% of the population of the department of Antioquia), and an equatorial-type climate (https://www.dane.gov.co [accessed on 31 July 2021]). The research was carried out in the Urabá region (Antioquia-Colombia). The sampling took place in five different municipalities (Place 1—Chigorodó: 7°40′11′′ N 76°40′53′′ O), (Place 2—Carepa: 7°45′29′′ N 76°39′19′′ O), (Place 3—Apartadó: 7°53′05′′ N 76°38′06′′ O), (Place 4—Turbo: 8°05′35′′ N 76°43′42′′ O), (Place 5—Necoclí: 8°25′33′′ N 76°47′02′′ O).

### 2.3. Capture of Bats

The bats were captured using mist nets of 2 m high with variable lengths of 6 and 12 m. The nets were placed in strategic areas near fruit trees and in areas of bat traffic after night observation. The captures were made during 4 continuous nights from 5:00 pm to midnight. The traps were checked every 30 min. Captured pregnant or lactating female bats were released immediately. These specimens were stored in cotton cloth bags until euthanasia and dissection. All captured animals were registered with a unique code. Additionally, the bat capture sites were georeferenced by GPS and the maps were generated using the environment and programming language R and packages (ggplot2, MappingGIS, sfMaps, spData, ggrepel, ggspatial, cowplot). 

### 2.4. Euthanasia of Captured Bats

The euthanasia process was carried out under the guidelines of AVMA Guidelines for the Euthanasia of Animals—2020. Initially the bats were sedated with 0.1 mL of 2% Xylazine; euthanasia was performed using a mixture of 390 milligrams Sodium Pentobarbital and 50 milligrams Sodium Diphenyl Hydantoin. The injection was performed intramuscularly in the pectoral region with insulin syringes. Dissection and collection of organs of interest were performed and the animal’s body was stored in 80% ethanol for conservation. Subsequently, bats were identified at the level of genus and species through morphological keys [40].

### 2.5. DNA Extraction

DNA extraction was performed with the commercial kit (Wizard DNA extraction kit, Promega^®^, Madison, WI, USA) according to the manufacturer’s recommendations for Gram negative bacteria.

### 2.6. PCR-16S Ribosomal Gene Conditions

Amplification of the 16S ribosomal gene by PCR was performed, as described previously, by Peláez et al. (2017) [41]. 

### 2.7. 16S Ribosomal Gene Sequencing from Kidneys of Bats

The amplification products were column-purified and sent to Macrogen^®^ Company (Seoul, Korea) for sequencing.

### 2.8. Phylogenetic Analysis 16S Ribosomal Gene

Phylogenetic analysis was performed, as described previously, by Peláez et al. (2017) [41].

## 3. Results

### 3.1. Places of Bat’s Capture

The investigation was carried out in the Urabá region (Antioquia-Colombia). The sampling was undertaken in five different municipalities (Chigorodó—43 captured bats), (Carepa—43 captured bats), (Apartadó—39 captured bats), (Turbo—40 captured bats), (Necoclí—41 captured bats). In total, 206 bats were captured. The map of the Urabá region and the exact location of the three sampling sites are shown in Figure 1.

### 3.2. Families, Genera and Species of Captured Bats

We captured 206 bats in the five municipalities of the Urabá region (Antioquia, Colombia). These bats were classified into three different families (Phyllostomidae, Molossidae, and Vespertilionidae), 10 different genera (*Artibeus*, *Carollia*, *Dermanura*, *Glossophaga*, *Sturnira*, *Molossus*, *Myotis*, *Uroderma*, *Rhogeessa*, *Phyllostomus*), and fifteen different species (*Artibeus*
*jamaicensis*, *A. lituratus*, *A. planirostris*, *Carollia brevicauda*, *C. castanea*, *C. perspicillata*, *Dermanura rava*, *Glossophaga soricina*, *Sturnira bakeri*, *Molossus molossus*, *Myotis caucensis*, *Uroderma convexum*, *Phyllostomus hastatus*, *P. discolor*), and one unidentified species belonging to the genus *Rhogeessa*. The genera, families and species are shown in Figure 2. These species have different feeding habits, such as frugivorous (60.19%), insectivores (17.47%), omnivore (1.45%), nectarivores (20.87%) (Table 1).

### 3.3. Detection of Leptospira spp. in Bats by Conventional PCR

We analyzed 206 bat kidneys by PCR by amplifying the 16S ribosomal gene for detection of *Leptospira* spp. Twenty individual bats were positive for *Leptospira* (20/206), obtaining a 9.7% of infected bats (Figure 3). Positive bats for *Leptospira* infection were found in the 5 municipalities studied (Chigorodó: 3 bats, Carepa: 2 bats, Apartadó: 3 bats, Turbo: 10 bats, and Necoclí: 2 bats). Additionally, 6 different species of bats were found to be infected: *Carollia perspicillata*, *Dermanura rava*, *Glossophaga soricina*, *Molossus molossus*, *Artibeus planirostris*, and *Uroderma convexum.* According to sex, 11 males (55%) and 9 females (45%) were found infected. Regarding feeding habits, 12 frugivores (60%), 6 nectarivores (30%), and 2 insectivores (10%) bats were found infected (Table 2).

### 3.4. Identification of Leptospira Species by Phylogenetic Analysis

Through the amplification, sequencing, and phylogenetic analysis of the 20 positive bat samples, the following *Leptospira* species were identified: *Leptospira borgpetersenii* (3/20–15%), *Leptospira alexanderi* (2/20–10%), *Leptospira noguchii* (6/20–30%), *Leptospira interrogans* (3/20–15%), and *Leptospira kirschneri* (6/20–30%). Results of the phylogenetic identification are shown in Figure 4.

### 3.5. Host-Pathogen Relationship between Bats and Leptospira

The host-pathogen association is as follows: *Leptospira borgpetersenii* infected 2 bats species (*Glossophaga soricina* and *Artibeus planirostris*), *Leptospira alexanderi* infected 2 bats species (*Uroderma convexum* and *Glossophaga soricina*), *Leptospira noguchii* infected 3 bats species (*Glossophaga soricina*, *Uroderma convexum*, and *Molossus molossus*), *Leptospira interrogans* infected 3 bats species (*Glossophaga soricina*, *Artibeus planirostris*, and *Uroderma convexum*) and *Leptospira kirschneri* infected 5 bats species (*Carollia perspicillata*, *Dermanura rava*, *Glossophaga soricina*, *Molossus molossus*, and *Artibeus planirostris*). The number of infected bats for each *Leptospira* species is shown in Table 3. Additionally, no renal infection was detected in 9 bat species (*A. jamaicensis*, *C. brevicauda*, *C. castanea*, *S. bakeri*, *A. lituratus*, *M. caucensis*, *P. hastatus*, *P. discolor*, and *Rhogeessa* sp.).

## 4. Discussion

Leptospirosis is a zoonotic disease that affects multiple animal reservoirs such as rodents [42], cattle [43], pigs [44], canines [45], capybaras [46], primates [47], turtles [48], sea lions [49], reptiles [50], bats [51], and other animals. Bats have gained great importance as efficient reservoirs and disseminators of *Leptospira* species for their biological attributes of hibernating [18], forming large colonies [19], migrating long distances [20], and having a long lifespan [21]. Although Neotropical bat species are not known to hibernate, that ability could favor the continuous maintenance of the bacteria in the host. Large colony formation facilitates transmission between different bats. Due to their ability to fly and migrate great distances, they could be an important bridge between urban, rural, and wild cycles of leptospirosis. Additionally, the longevity of bats could favor the dispersion of the bacteria through urine for prolonged periods of time into different environments and animals.

Worldwide, bats infected with *Leptospira* have been reported in at least 16 countries, with 50 different species of infected bats, and four *Leptospira* spp. as causative agents of the infection (*L. interrogans*, *L. borgpetersenii*, *L. kirschneri*, *L. fainei*) [22]. In Colombia the situation is no different, two studies reported bats infected with *Leptospira*. In the first study, 2 bats species (*Eumops nanus, Lonchophylla fornicata*) captured in schools belonging to the municipality of Sincelejo, Sucre, were found positive for *Leptospira* infection [39]. In the second study, 6 bats species (*C. perspicillata, G. soricina, U. convexum, D. phaeotis, Desmodus rotundus, Lophostoma silvicola*) deposited in the Museum of Natural History of the Pontificia Universidad Javeriana, in Bogotá, from dry forests of Córdoba department, were analyzed by PCR and their kidneys were positive for *Leptospira* infection [40]. Few studies have been conducted in Colombia; it is necessary to carry out the identification and characterization of infected bats with *Leptospira* in other regions of the country to decipher the biological role played by bats in the transmission cycle of leptospirosis in Colombia. 

The objective of our investigation was to detect and identify *Leptospira* species infecting bats in the Antioquia Department. In the present study, 206 bats were captured, which were identified as belonging to 15 different species. The study found that 40% of bat species were infected with *Leptospira*, while 60% of the species were not infected. The three most abundant species were *Artibeus planirostris* (26.69%), *Glossophaga soricina* (20.87%), and *Molossus molossus* (12.62%). In these most abundant species, at least one infected bat was found, suggesting a large number of infected bats at sampling sites. Interestingly, infected bats were also found in species with high, medium, and low abundance, which indicates that infection is independent of the abundance of bat populations. Also, 9 bats species of medium and low abundance were not infected, representing 60% of the species analyzed (*A.*
*jamaicensis*, *A. lituratus*, *C. brevicauda*, *C. castanea*, *S. bakeri*, *M. caucensis*, *P. hastatus*, *P. discolor*, and *Rhogeessa* sp.) and 90.3% of the individuals analyzed (186/206). The absence of infection in these species may be due to the small number of individuals captured in the sampling process or a mechanism of natural resistance to infection by these bats.

Regarding feeding habits, the infected bats presented feeding habits such as insectivores, frugivores, and nectarivores; meanwhile, the uninfected bats presented eating habits such as frugivores, insectivores, and omnivores. Being omnivores the only difference between infected and uninfected bats, respectively. Another important finding in this study was the identification of five pathogenic *Leptospira* species infecting 40% of the species of captured bats *(Leptospira borgpetersenii*, *Leptospira alexanderi*, *Leptospira noguchii*, *Leptospira interrogans*, and *Leptospira kirschneri).* This result highlights the importance of bats as important reservoir hosts and disseminators of multiple pathogenic *Leptospira* species in the Urabá region (Antioquia-Colombia). It is important to highlight that 6/20 infected bats correspond to *Leptospira borgpetersenii* and *Leptospira noguchii,* which emphasize the role of bats in maintaining these species of *Leptospira* in a wild. Additionally, these five *Leptospira* species are found in the P1 taxonomic group, which contains the most virulent of the human *Leptospira* species [4,5]. The infection rates for the different species were *Leptospira borgpetersenii* (3/20–15%), *Leptospira alexanderi* (2/20–10%), *Leptospira noguchii* (6/20–30%), *Leptospira interrogans* (3/20–15%), and *Leptospira kirschneri* (6/20–30%). This is the first report in which these five pathogenic *Leptospira* species are identified infecting bats in the wild (Antioquia-Colombia). Additionally, these findings suggest the importance of bats in the dispersion and circulation of pathogenic *Leptospira* species into the environment. Given their ability to fly long distances, bats could serve as a bridge between wild and urban cycles of leptospirosis. Bats have been identified worldwide as an important reservoir of different *Leptospira* spp. (*L. interrogans*, *L. borgpetersenii*, *L. kirschneri*, *L. fainei*) [22]. In this study, *Leptospira alexanderi* and *Leptospira noguchii* are reported for the first time infecting bats. With respect to this host-pathogen relationship, it is noted that one species of *Leptospira* can infect multiple species of bats without being influenced by their feeding habits or population density, suggesting the presence of the bacteria in multiple environments.

The positive bats for *Leptospira* infection in the Urabá region correspond to six species in six genera and two families, Phyllostomidae and Molossidae. Almost all the phyllostomid bats are frugivorous, except for *G. soricina*, which is a nectarivorous species. Conversely, *M. molossus* is an insectivorous bat that feeds on small insects on the fly [52]. In contrast, *C. perspicillata*, *A. planirostris*, *G. soricina*, and *U. convexum* were relatively common in our netting effort; we found *M. molossus* in large numbers mostly because we netted close to their roosts near an old building. It appears that periurban areas where we netted still maintain a vegetation structure that allows these bat species to find roosting sites and food for their persistence [53,54,55], but also it enforces the fact that these species show high tolerance to landscape transformation (such as forest fragmentation [56]. Although two of these bats, *G. soricine* and *M. molossus*, have been found roosting nearby people’s houses [57], direct interactions with people are rarely reported. Bats roosting in human spaces present risks because these are the places where bats conduct most of their activities, which may include deposition of urine and droppings that may carry the pathogen and contaminate water or food sources [39,58].

In this study, we showed that bats in the Urabá region (Antioquia-Colombia) are important reservoirs and disseminators of pathogenic *Leptospira* species. With changing habitats due to man-made interventions, bats are becoming a significant reservoir of many zoonotic pathogens. These findings will help us in understanding the role played by bats in the infectious cycle of leptospirosis and help in the implementation of better prevention and control measures for leptospirosis in our country.

## Figures and Tables

**Figure 1 microorganisms-09-01897-f001:**
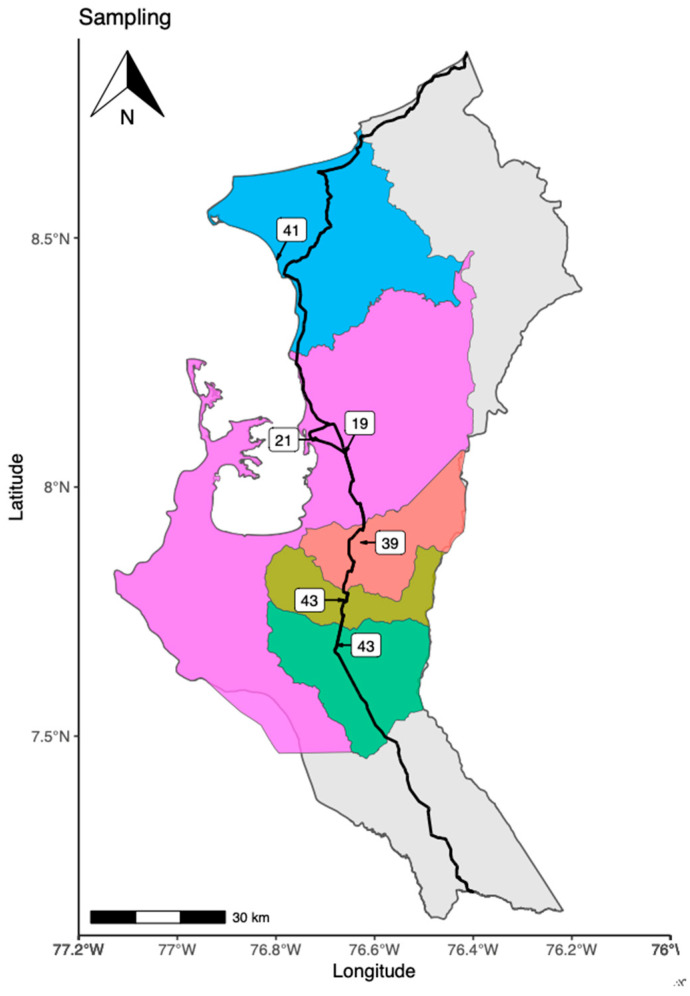
The geographical location of capture sites. The map shows the geographical location of the five municipalities that were used to capture the 206 bats used in this study (blue-Necoclí, purple-Turbo, orange-Apartadó, dark green-Carepa and light green-Chigorodó). These capture sites are located in the Urabá region (Antioquia-Colombia). The map was generated using the environment and programming language R and packages (ggplot2, MappingGIS, sfMaps, spData, ggrepel, ggspatial, cowplot).

**Figure 2 microorganisms-09-01897-f002:**
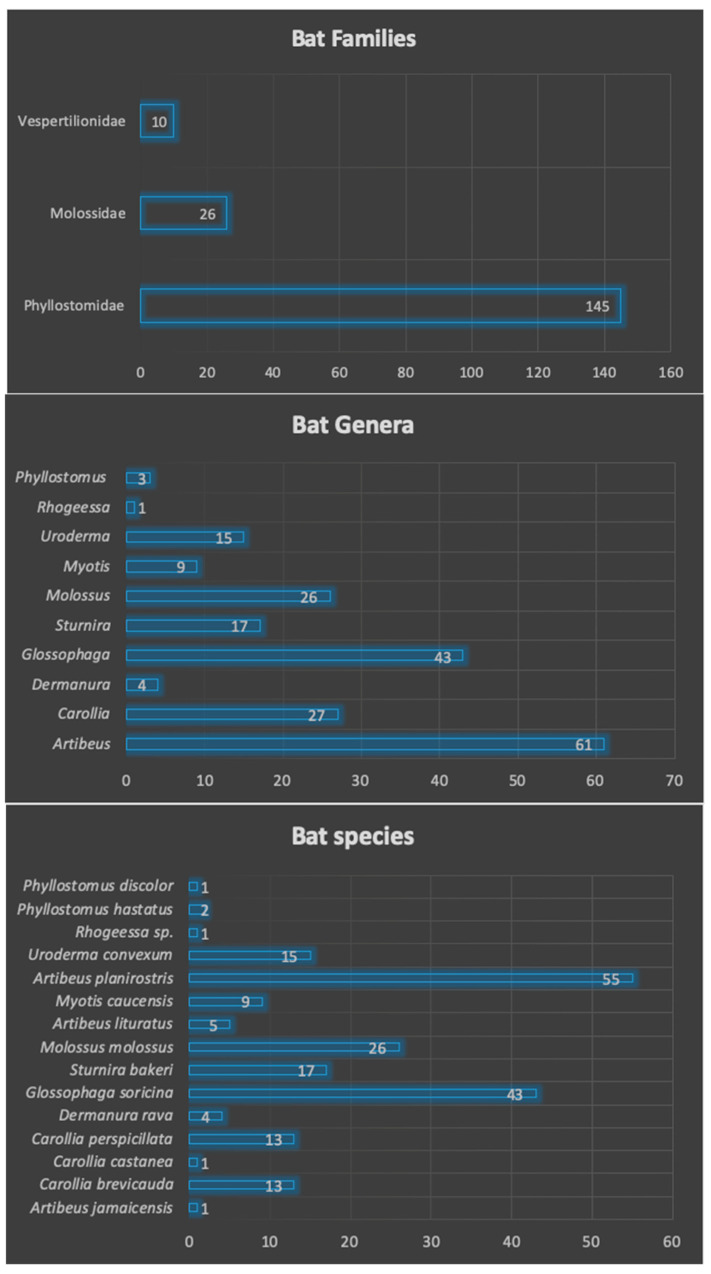
Diversity and abundance of bats captured in the five municipalities of the Urabá region. Figure 2 shows the 3 families, 10 genera, and 15 species of bats that were captured in the five sampling areas. The number of individuals for each taxonomic group classification are also indicated.

**Figure 3 microorganisms-09-01897-f003:**
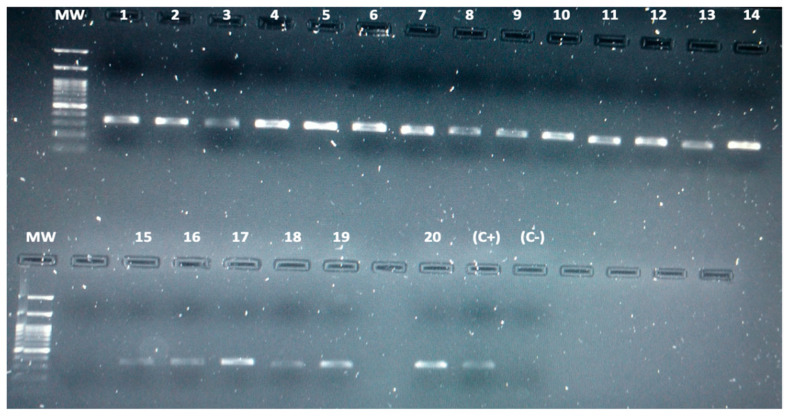
Molecular detection of bats naturally infected with *Leptospira*. The figure shows a 1% agarose gel with the amplification products of 20 bats infected with *Leptospira* spp. The band (331 base pair) corresponding to a fragment of the 16S ribosomal gene. A 100 base pair molecular weight markers were used. Additionally, a positive control (C+: *Leptospira interrogans*) and a negative control (C-: PCR reagents without DNA) were used in all reactions.

**Figure 4 microorganisms-09-01897-f004:**
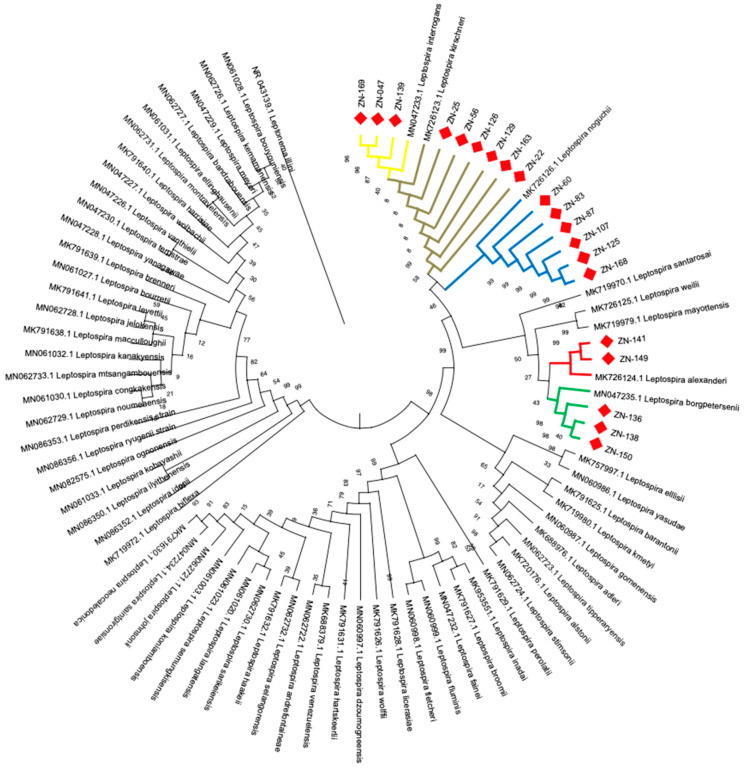
Identification of *Leptospira* species infecting bats by phylogenetic analysis of the 16S ribosomal gene. Phylogenetic reconstruction of the 16S ribosomal gene of the genus *Leptospira* is shown. Red diamonds represent the bats infected with *Leptospira* spp. in this study. *Leptospira borgpetersenii*, *Leptospira alexanderi*, *Leptospira noguchii*, *Leptospira interrogans*, and *Leptospira kirschneri* were the *Leptospira* species found infecting this bat population.

**Table 1 microorganisms-09-01897-t001:** Diversity of bats captured in the study. This table shows information about the species, number, percentage, frequency and feeding habits of the 206 bats captured.

Species	Number	Percentage (%)	Frequency	Feeding Habits
*Artibeus jamaicensis*	1	0.49%	0.005	frugivore
*Carollia brevicauda*	13	6.31%	0.063	frugivore
*Carollia castanea*	1	0.49%	0.005	frugivore
*Carollia perspicillata*	13	6.31%	0.063	frugivore
*Dermanura rava*	4	1.94%	0.019	frugivore
*Glossophaga soricina*	43	20.87%	0.209	nectarivore
*Sturnira bakeri*	17	8.25%	0.083	frugivore
*Molossus molossus*	26	12.62%	0.126	insectivore
*Artibeus lituratus*	5	2.43%	0.024	frugivore
*Myotis caucensis*	9	4.37%	0.044	insectivore
*Artibeus planirostris*	55	26.70%	0.267	frugivore
*Uroderma convexum*	15	7.28%	0.073	frugivore
*Rhogeessa* sp.	1	0.49%	0.005	insectivore
*Phyllostomus hastatus*	2	0.97%	0.010	omnivore
*Phyllostomus discolor*	1	0.49%	0.005	omnivore
TOTAL	206	100%	1	

**Table 2 microorganisms-09-01897-t002:** Natural infection of bats with different *Leptospira* species. This table shows the code of the positive samples, *Leptospira* species identified by amplification of the 16S ribosomal gene, bat species infected, and the municipality from which the sampling area originated.

Code	Phylogenetic Identification (16S Ribosomal Gene)	Infected Species	Feeding Habits	Gender	Municipality
ZM-022	*Leptospira kirschneri*	*Carollia perspicillata*	Frugivore	Female	Carepa
ZM-025	*Leptospira kirschneri*	*Dermanura rava*	Frugivore	Male	Carepa
ZM-047	*Leptospira interrogans*	*Glossophaga soricina*	Nectarivore	Female	Apartadó
ZM-056	*Leptospira kirschneri*	*Glossophaga soricina*	Nectarivore	Male	Apartadó
ZM-060	*Leptospira noguchii*	*Glossophaga soricina*	Nectarivore	Female	Apartadó
ZN-083	*Leptospira noguchii*	*Uroderma convexum*	Frugivore	Male	Chigorodó
ZN-087	*Leptospira noguchii*	*Uroderma convexum*	Frugivore	Male	Chigorodó
ZN-107	*Leptospira noguchii*	*Uroderma convexum*	Frugivore	Female	Chigorodó
ZN-125	*Leptospira noguchii*	*Molossus molossus*	Insectivore	Female	Turbo
ZN-126	*Leptospira kirschneri*	*Molossus molossus*	Insectivore	Male	Turbo
ZN-129	*Leptospira kirschneri*	*Artibeus planirostris*	Frugivore	Male	Turbo
ZN-136	*Leptospira borgpetersenii*	*Glossophaga soricina*	Nectarivore	Female	Turbo
ZN-138	*Leptospira borgpetersenii*	*Glossophaga soricina*	Nectarivore	Female	Turbo
ZN-139	*Leptospira interrogans*	*Artibeus planirostris*	Frugivore	Male	Turbo
ZN-141	*Leptospira alexanderi*	*Uroderma convexum*	Frugivore	Female	Turbo
ZN-149	*Leptospira alexanderi*	*Glossophaga soricina*	Nectarivore	Male	Turbo
ZN-150	*Leptospira borgpetersenii*	*Artibeus planirostris*	Frugivore	Male	Turbo
ZN-163	*Leptospira kirschneri*	*Artibeus planirostris*	Frugivore	Male	Turbo
ZN-168	*Leptospira noguchii*	*Uroderma convexum*	Frugivore	Male	Necoclí
ZN-169	*Leptospira interrogans*	*Uroderma convexum*	Frugivore	Female	Necoclí

**Table 3 microorganisms-09-01897-t003:** Natural infection of bats with different *Leptospira* species. The table shows the host-pathogen relationship between 6 *Leptospira* species and 6 bats species susceptible to infection. The number of bats infected by each *Leptospira* species is shown in parentheses.

*Leptospira* Species	Infected Bat Species	Infected Bats
*Leptospira borgpetersenii*	*Glossophaga soricina* (2) *Artibeus planirostris* (1)	3
*Leptospira alexanderi*	*Uroderma convexum* (1) *Glossophaga soricina* (1)	2
*Leptospira noguchii*	*Glossophaga soricina* (1) *Uroderma convexum* (4) *Molossus molossus* (1)	6
*Leptospira interrogans*	*Glossophaga soricina* (1) *Artibeus planirostris* (1) *Uroderma convexum* (1)	3
*Leptospira kirschneri*	*Carollia perspicillata* (1) *Dermanura rava* (1) *Glossophaga soricina* (1) *Molossus molossus* (1) *Artibeus planirostris* (2)	6
		Total: 20

## Data Availability

The genome sequences of the strains sequenced in this study have been deposited in GenBank under accession numbers MZ853085-MZ853104.

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
