# Peer review of "High Diversity of Leptospira Species Infecting Bats Captured in the Urabá Region (Antioquia-Colombia)"

_microorganisms, 2021, doi:10.3390/microorganisms9091897_

Round 1

Reviewer 1 Report

1) The introduction is concise and gives enough information about the studied problem worldwide. Moreover, the importance of the problem is emphasized.     

2) About the part “Methods”:

The Study Design is appropriate for achievement of the aim of the study. All used methods are relevant to the design of the study and are precisely described.

3) In the section “Results” in five subparts are mentioned all findings about the studied problem. The order of the subparts follows the logical structure of the study. There are four figures and three tables that support the results.

4) In the section “Discussion” the authors had shown that “worldwide, bats infected with Leptospira have been reported in at least 16 countries, with 50 different species of infected bats, and 4 Leptospira spp. as causative agents of the infection (L. interrogans, L. borgpetersenii, L. kirschneri, L. fainei)”. In the same time, they mentioned the studies in Colombia about the discussed problem. As the authors mentioned, an “important finding in this study was the identification of five pathogenic Leptospira species infecting 37,5% of the species of captured bats (L. borgpetersenii, L. alexanderi, L. noguchii, L. interrogans, and L. kirschneri). This result highlights the importance of bats as important reservoir hosts and disseminator of multiple pathogenic Leptospira species in the Urabá region (AntioquiaColombia). It is important to highlight that 6/20 infected bats correspond to L. borgpetersenii and L. noguchii which highlights the importance of bats in maintaining these species of Leptospira in a wild. Additionally, these five Leptospira species are found in the P1 taxonomic group, which contains the most virulent of the human Leptospira species. This is the first report in which these 5 pathogenic Leptospira species are identified infecting bats in the wild (Antioquia-Colombia). Additionally, these findings suggest the importance of bats in the dispersion of pathogenic Leptospira species into the environment. Given their ability to fly long distances, bats could serve as a bridge between wild and urban cycles of leptospirosis.”

5) The conclusions are completely supported by the results of the study.    

6) In the list of References are included 59 sources of information (11 of them from last five years – 19%). The little number of recent articles is not disadvantage, because the studied problem is new. The articles are listed in order of mentioning in the text and all of them are cited correctly.

            At the end of my notes I want to say that this study gives valuable information about the role of the bats as a reservoir of Leptospira and deserves to be published. It will be of strong interest to the readers of the journal. 

My final recommendation is to accept the manuscript in present form. 

Author Response

There were no comments to address but we want to thank reviewer 1 for taking time to review our manuscript.

Reviewer 2 Report

This is well-written and I have no criticism of the methods, results and conclusions. There are some very minor grammar corrections to be made by the editor. But, for the most part, this already reads like a final manuscript.

I have only one criticism of the paper, but this should warrant checking other references. In the introduction, the authors state that leptospirosis has been found in bats around the world, including the USA. Their reference (#36) is Harkin, Hays, Davis and Moore. But that study did not identify leptospirosis in any bat (all Eptesicus fuscus). I am not aware of any study that documents leptospirosis in bats in the US (one case report inferred but did not prove). 

Author Response

We want to thank reviewer 2 for bringing to our attention reference #36. While this work was performed in the USA, no Leptospira was detected in bats. We have made this change in the revised manuscript.